# Clinical Perspectives of Single-Cell RNA Sequencing

**DOI:** 10.3390/biom11081161

**Published:** 2021-08-06

**Authors:** Nayoung Kim, Hye Hyeon Eum, Hae-Ock Lee

**Affiliations:** 1Department of Microbiology, College of Medicine, The Catholic University of Korea, Seoul 06591, Korea; wwkd0324@gmail.com (N.K.); hheum1107@gmail.com (H.H.E.); 2Department of Biomedicine and Health Sciences, Graduate School, The Catholic University of Korea, Seoul 06591, Korea

**Keywords:** single-cell genomics, single-cell RNA sequencing, treatment response, patient stratification, clinical decision

## Abstract

The ability of single-cell genomics to resolve cellular heterogeneity is highly appreciated in cancer and is being exploited for precision medicine. In the recent decade, we have witnessed the incorporation of cancer genomics into the clinical decision-making process for molecular-targeted therapies. Compared with conventional genomics, which primarily focuses on the specific and sensitive detection of the molecular targets, single-cell genomics addresses intratumoral heterogeneity and the microenvironmental components impacting the treatment response and resistance. As an exploratory tool, single-cell genomics provides an unprecedented opportunity to improve the diagnosis, monitoring, and treatment of cancer. The results obtained upon employing bulk cancer genomics indicate that single-cell genomics is at an early stage with respect to exploration of clinical relevance and requires further innovations to become a widely utilized technology in the clinic.

## 1. Introduction

Cells are the basic units of all living organisms. Single-cell genomics has ushered in a new era in biology by employing genomics to define cellular identity and states. Genomic data of individual cells provide the status of thousands of genes or regulatory elements simultaneously. For instance, data pertaining to the transcriptome and the epigenome (chromatin openness) can be obtained through single-cell RNA sequencing (scRNA-seq) and single-cell ATAC-seq (assay for transposase-accessible chromatin with sequencing), respectively. As cellular identity and function are closely linked to the spatial and temporal positions of cells, genomic data complement the traditional cell type definition that takes into consideration of the anatomical, morphological, and developmental traits. Massively parallel genomic analyses recapitulate dynamic cellular states and therefore, single-cell genomics provides high-resolution criteria for determining the cellular identities in health and disease conditions.

Cancer is defined as an abnormal growth of cells characterized by uncontrolled division and the ability of these cells to invade and destroy normal tissue architectures. From a genomic perspective, cancer is a disease that arises as a result of genomic instability, a phenomenon that disrupts the normal genomic program in healthy cells within the tissue context. Indeed, single-cell transcriptome analysis has demonstrated the separation of cancer cells from normal counterpart cells with the cancer cells positioned in distinct clusters in each patient while non-malignant cells show multi-patient clustering, suggesting a gross deviation from the normal genomic programs [1,2] (Figure 1A–C). The transcriptional clusters may—in part—represent specific genetic alterations or transcriptional modules associated with tumor progression, metastasis, and response, and resistance to a specific treatment [1,3,4,5]. In addition to the genomic characterization of tumor cells, single-cell genomics allows for the cellular dissection and the characterization of the tumor microenvironment, which influences tumor cell behavior and the effect of anti-cancer therapies directed at stromal and immune cells [2] (Figure 1D).

In clinical practice, the benefits associated with the implementation of single-cell resolution cancer genomics are uncertain. Here, we will review the clinical implementation of conventional cancer genomics and examine clinical cases and areas that are addressed by single-cell genomics, and then discuss whether and how this technology will promote the development of better strategies to diagnose, monitor, and treat cancer.

## 2. Brief Overview of Cancer Genomics in Precision Oncology

There were quite some hurdles to utilizing cancer genome data to make clinical decisions regarding the treatment choices. Molecular target discoveries that aid platform development for clinical implementation would enable us to obtain a better understanding whether and how the single-cell genomics data can be incorporated into the clinical decision-making process (Table 1).

### 2.1. Molecular Target Discoveries and NGS Panel Sequencing

Imatinib (Gleevec)—targeting BCR-ABL in chronic myeloid leukemia—is one of the most successful molecular-targeted therapies for genetic aberrations in cancer [7]. It took 30 years from the discovery of the truncated chromosome 22—Philadelphia chromosome—to the identification of the *BCR-ABL* gene fusion and another 10 years for the clinical trial on the tyrosine kinase inhibitor imatinib [8]. Next generation sequencing (NGS) enabled the analysis of the entire human genome and influenced the launching of cancer genome projects such as The Cancer Genome Atlas (TCGA)—in 2005—and the International Cancer Genome Consortium (ICGC) in 2008 [9], which led the community efforts to obtain a comprehensive understanding of the genomic alterations underlying all major cancers. Starting from the first markup study on the glioblastoma genome [10], TCGA has provided insights into the genomic landscapes of over 33 cancer types [11] and ICGC has provided insights into the genomic landscapes of over 50 cancer types in less than 15 years. It suffices to say that now we have a genomic draft for major cancer types. The draft, which focused on the protein-coding regions and expanded to the non-coding areas of the human genome, remarkably accelerated the discovery of genetic aberrations in cancer.

While whole genome, exome, or transcriptome sequencing provide more extensive information about genomic alterations in cancer, currently only gene panel sequencing is employed in clinical settings [11,12]. Oncomine Dx Target Test, FoundationOne CDx, and MSK-IMPACT are the representative US FDA-approved NGS panel platforms covering different ranges of point mutations, gene amplifications, translocations, and complex genomic features such as tumor mutational burden (TMB) or homologous recombination deficiency (HRD) that guide molecular-targeted therapies in lung cancer or multiple cancer types [13]. The tangent from the emergence of cancer genomics to the clinical implementation of panel sequencing indicates the requirement of two parallel areas of research, i.e., one linking genomic alterations to treatment response [14,15] and the other involving the formulation of efficient strategies for detecting clinically significant alterations [16].

Association of genomic alterations with treatment response in cancer is categorized in multiple tiers [14], with druggable mutations used in companion diagnostics (CDx) at the highest level, cancer mutations with evidence of clinical significance at the second level, and cancer mutations with potential clinical significance at the third level. For example, FoundationOne CDx in the current form detects genetic aberrations in 324 genes and three genomic features, i.e., microsatellite instability (MSI), TMB, and HRD status [17]. Among these, CDx indications include non-small cell lung cancer (NSCLC) with *EGFR* exon 19 deletions and *EGFR* exon 21 L858R alterations (for afatinib, gefitinib, or erlotinib); *EGFR* exon 20 T790M for osimertinib; *ALK* rearrangements (for alectinib, crizotinib, or ceritinib); and *BRAF* V600E (for dabrafenib in combination with trametinib). Additional indications include melanoma with *BRAF* 600E and V600K alterations, breast cancer with *ERBB2* (HER2) amplification, colorectal cancer with *KRAS* wild-type (absence of mutations in codons 12 and 13, or exons two, three, and four) and *NRAS* wild-type, and ovarian cancer with *BRCA1/2* alterations. TMB (>10 mutations per megabase) can be used as a marker for the outcomes of pembrolizumab immune checkpoint therapy in solid tumors. CDx level genomic alterations are relatively few among all the detected genetic aberrations in a given assay, but are slowly expanding with the emergence of stronger evidence of clinical significance, as seen in the case of *NTRK* gene fusion [18].

### 2.2. Gene-Expression-Based Patient Stratification

Another example of the involvement of genomic datasets in the clinical decision-making process is prognostic testing. Oncotype Dx and Mammaprint are representative assays used in the recurrence risk assessment of early-stage breast cancer [19]. Oncotype Dx provides a recurrence score obtained from RT-PCR-based expression data of 21 genes, and Mammaprint reveals the risk of distant recurrence 10 years after diagnosis based on a microarray-based gene expression score of 70 genes [20]. These assays are recommended in clinical guidelines, such as the National Comprehensive Cancer Network (NCCN) and the American Society of Clinical Oncology (ASCO), as prognostic and therapy-predictive (Oncotype Dx only) assays complementing other clinical data. In multiple myeloma, MyPRS [21] and SKY-92 [22] provide microarray-based gene expression signature scores for risk stratification. However, these remain investigational, and their clinical implementation requires further validation. Though development of gene-expression-based complementary diagnostics has relied on the accumulation of microarray data with clinical annotations over many years, RNA sequencing has now emerged as the preferred method for high-throughput gene expression analysis. By comparison, RT-PCR is mostly employed as an assay for evaluating the expression of a small number of genes and is comparable to the panel sequencing strategy that is used for the efficient detection of selective DNA aberrations.

## 3. Tumor Heterogeneity Is Responsible for Treatment Resistance in Precision Oncology

Heterogeneous tumor cell populations—with variations in the proliferative capacity, tumor antigen expression, immune cell activation, or other characteristics affecting tumor growth and the response to anti-cancer drugs—are present within a single tumor. Intratumoral heterogeneity, investigated based on protein marker expression in tissues or isolated tumor populations—before the genomics era—has long been suspected to be responsible for the development of treatment resistance. Genomic analyses nowadays allow the systemic assessment of tumor cell properties, and multi-regional or single-cell genomics elaborate on the degree of intratumoral heterogeneity and its effects.

### 3.1. Tumor Heterogeneity Fine-Tuning Molecular-Targeted Therapies

As conditions associated with genomic instability, tumors evolve to generate multiple clones with discrete genomic profiles [23]. Multi-regional and longitudinal genomic studies on cancer have demonstrated the presence of genetic clones with different mutations, copy number alterations, and multiple epigenome and transcriptome profiles in a single patient [24,25]. Intratumoral heterogeneity poses a serious challenge in genome-based molecular-targeted therapy, as pre-existing non-target populations persist and result in disease relapse. To avoid the incomplete targeting of subset populations, treatments directed at truncal mutations shared by all tumor cells or drug combinations targeting all subclones can be adopted [26]. This approach involves the inclusive sampling of tumor tissues representing the whole tumor, which is difficult to achieve with biopsy specimens. Furthermore, tumor evolution has a much broader scope than that of pre-existing non-target clones. In a recent series of publications, the TRAcking non-small cell lung Cancer Evolution through therapy (TRACERx) consortium demonstrated spatial tumor heterogeneity in lung cancer arising from genetic evolution, transcriptomic diversification, and interplay with the immune microenvironment [27,28,29,30,31,32,33].

While multi-regional or multi-temporal sequencing is mainly used for the phylogenetic reconstruction of tumor evolution, tumor subclonal heterogeneity can be estimated within a single tumor sample based on the variant allele frequency of single nucleotide variants (SNVs) and large copy number alterations (CNAs) [34]. Tumor subclonal heterogeneity, which can be estimated from bulk sequencing data, is associated with more aggressive tumor growth, metastasis, and treatment resistance [25,35,36]. DNA sequencing at the single cell or the single nucleus levels also enables the reconstruction of subclonal structures in cancer, with both advantages and disadvantages compared to bulk analysis [37]. Single-cell or nuclear CNA overcomes the issues associated with ambiguities in tumor purity or ploidy as well as insensitivities to rare clones, if successfully captured, in bulk samples. However, the small number of assayed cells may not represent the diverse subclonal populations in the whole tumor. Uneven and low genome coverage due to the limited starting material and the extensive amplification also introduce errors in the estimation of CNA in single cells [37]. While ongoing technical advancement continues to increase the resolution and the scope of single-cell/nucleus-level DNA analysis in the reconstruction of tumor evolutionary paths [38], these limitations yield a bias and an underestimation of subclonal heterogeneity at single-cell resolution. Thus, in the estimation of genetic heterogeneity, bulk-level analyses are likely to provide reliable information regarding the complex subclonal structures that may affect the efficacy of molecular-targeted therapies.

Detection of genetic aberrations in circulating tumor cells (CTCs) or in tumor cells of minimal/measurable residual disease (MRD) could provide an early window for molecular target selection ahead of metastasis or relapse [39]. The frequencies of CTCs in solid tumors or in the residual tumor load during remission are extremely low, and bulk genomic analysis often fails to capture the tumor cell signals. Enrichment of EPCAM+CD45-cells and subsequent single-cell DNA sequencing, rather than simple enumeration, could provide information regarding the genetic characteristics of CTCs with a high metastatic potential in solid tumors. In acute myeloid leukemia (AML), tumor clones can be detected in 80% of patients during remission using single-cell sequencing [40]. While single-cell level DNA sequencing allows for more comprehensive genetic profiling of specific tumor clones, capturing rare tumor clones requires large numbers of single-cell sampling. Thus, the development of an efficient target cell capture platform—in terms of throughput and cost—is required for the practical use of single-cell genomics for monitoring the CTCs or residual tumor load.

### 3.2. Gene Expression Heterogeneity Associated with Prognosis, Treatment, and Patient Stratification

Unlike the assessment of genotype diversity at the DNA level, intratumoral heterogeneity assessment based on gene expression data requires analyses at a single-cell resolution due to its quantitative nature. At the tissue level, gene expression data have been used for tumor subtype assessment and patient stratification in multiple cancer types [41,42,43]. As scRNA-seq data dissect gene expression heterogeneity at the cellular level, the tumor subtype classification gets updated to distinguish tumor signatures and stromal characteristics surrounding the tumor [44] (Figure 2 and Table 2).

First, single tumor sites were frequently populated by multiple tumor subtype cells, implying compound tumor cell behavior. In glioblastoma, individual tumor cells in a single patient can be classified into diverse subtypes, such as proneural, neural, classical, and/or mesenchymal types, as well as hybrid cell types [1]. This multiple subtype composition and hybrid status suggested a variation in the susceptibility of tumor cells to conventional and molecular-targeted therapies and underscored their plasticity as well. In addition, in breast cancer, the presence of the rare triple-negative breast cancer (TNBC) subtype cells in estrogen receptor (ER)-positive tumors indicated the existence of pre-existing tumor clones with intrinsic resistance to hormone therapy [53]. Second, tissue-level mesenchymal subtypes associated with poor patient survival reflect the overabundance of cancer-associated fibroblasts (CAFs) in colorectal and ovarian cancers [49,50,54]. In these tumors, conventional gene expression patterns signifying an epithelial–mesenchymal transition (EMT) were found to originate from CAFs rather than from tumor cells. In head and neck squamous cell carcinoma, partial EMT programs that are different from the conventional EMT signatures have been identified [52]. Identification of transcriptional programs governing the epithelial tumor cell-specific EMT could help redefine the regulatory pathways of metastasis and may reveal new strategies to control the metastatic phenotype. Third, the gene expression phenotype of tumor cells could be linked to molecular-targeted therapies and drug combinations. The presence of two tumor subpopulations with mutually exclusive EGFR or SRC signaling pathways facilitated the prediction of drug combinations, for which the efficacy was validated using a patient-derived tumor xenograft mouse model [5]. In addition, tumor cell-specific expression of a drug target may reinforce the utilization of genome-based molecular-targeted therapy [55]. Fourth, immune cell profiling demonstrated that T cell or myeloid cell phenotypes influenced the success of immune checkpoint therapies. In the past decade, remarkable responses to immune checkpoint therapy (ICT) using therapeutic agents such as programmed death-1/programmed death ligand-1 (PD-1/PD-L1) and cytotoxic T lymphocyte antigen-4 (CTLA4) inhibitors have been reported to have a huge impact on the management of cancer and development of cancer immunotherapies. Melanoma and lung cancer were the first tumor types against which ICT had shown efficacy; however, the overall response rate in these types was found to be less than 20%. Patient selection based on high PD-L1 expression—identified by immunohistochemistry or high TMB (or tumor neoantigen)—has increased the response rate to 40–60% [56]. As PD-L1 expression and high TMB resulted in the recapitulation of the immune cell-rich phenotype, immune profiling of the peripheral blood or tumor-infiltrating leukocytes (TILs) also predicted the response to ICT at some levels [57]. Nonetheless, this prediction is insufficient, and improved prediction methods could increase the success rate of ICT and aid the development of new strategies targeted at promoting tumor-directed immunity. In the single-cell genomic approach, immune cell profiling can be achieved at a much higher resolution, covering both cell types and cellular states (Table 3).

Comparison between responders and non-responders provides critical information regarding patient stratification and the mechanism of action of ICT [69]. Characterization of post-treatment samples helps define the immune cell dynamics and state transitions [70]. Overall, effector-memory CD8+ T cells are known to be associated with a positive response to ICT [61,64]. Exhausted phenotype T cells, likely including both activated and exhausted populations, have been reported to predict a positive ICT response in melanoma, NSCLC, and basal cell carcinoma [58,59,60,61,62,65,71,72]. In a more recent study, Transcription Factor 7 (TCF7)—but not *PDCD1* (PD-1 gene)—expression, was found to be a positive predictor of tumor regression in melanoma [63]. In MSI-high gastric cancer, a diverse T cell receptor (TCR) repertoire and the presence of early-to-intermediate activation status T cells in tumor tissues were found to be associated with an ICT response [68]. As T cell activation and exhaustion are confounding processes, exhaustion markers are good indicators of the presence of tumor-directed immunity; however, terminal differentiation of collective tumor-specific T cells in the exhausted state may adversely influence the ICT response. Single-cell genomics is a robust tool for defining the dynamic balance between T cell activation and exhaustion.

## 4. Multimodal and Deconvolution Approaches Facilitate the Biological Interpretations and Clinical Applications of Single-Cell Genomic Data

Among many single-cell genomic tools, transcriptome analysis is most frequently used to evaluate intratumoral heterogeneity due to the scalability of technology and the feasibility of phenotypic translation. Several attributes need to be considered to exploit the biological and clinical interpretations of single-cell transcriptome data (Figure 3).

Both genetic and non-genetic mechanisms are likely to influence heterogeneity at the level of gene expression. Heterogeneity arising from tumor-intrinsic genetic evolution or non-genetic alterations triggered by microenvironmental factors is associated with distinct indications that enable the heterogeneity-associated treatment resistance to be overcome, depending on the source. Thus, linking genetic data with gene expression heterogeneity is important to determine how to use the information for therapeutic advancement. Another important aspect of single-cell transcriptome analysis is the uncertainty in the number of clusters or subpopulations. Defining biologically meaningful clusters requires prior knowledge of gene expression obtained using well-isolated population data with functional corroboration. As the isolation procedures rely on a few surface protein markers, it is often the protein expression profile that defines a functional subpopulation. Therefore, the inference of transcriptome-based clusters needs to be validated by the simultaneous measurement of corresponding surface markers and gene expression. Finally, the use of single-cell RNA sequencing has practical limitations in terms of the cost and the associated sampling breadth. Though the cost has dropped substantially—and will continue to decrease—and the sampling breadth has increased, the output cell numbers in most studies remain below 3000 cells per sample, which may not be enough to cover extensive heterogeneity in complex tissues. Deconvolution methods using bulk genomic data—both at the genetic and the gene expression levels—may overcome the limitations associated with cost and sampling and could also provide an opportunity to treat a large number of patients besides offering substantial clinical annotations.

### 4.1. Linking Genetic and Gene Expression Data

Combining single-cell DNA and gene expression data may reveal tumor heterogeneity that could result in the translation of genetic evolution into transcriptional phenotypes. Multi-omics data generation methods [73] such as G&Tseq [74] and Trioseq [75]. Alternatively, genetic features, such as mutations and CNA, can be measured or inferred from scRNA-seq data (Table 4).

Detection of SNVs in single-cell RNA sequencing data is hampered by false negatives and positives caused by massive dropouts and amplification errors, respectively. Statistical methods may improve the detection accuracy to some extent [82,83]. Nonetheless, many single-cell RNA sequencing data do not cover full-length transcripts, and sequence variation analysis at specific sites remains an inefficient method to determine the clonality of tumor cells, even with full-length data. Inference of DNA CNAs from quantitative chromosomal gene expression data could provide a more robust method to identify genetic clones with large CNAs [76,77,78,79,80]. The supplementary use of single nucleotide polymorphisms (SNPs) increases the resolution of copy number inference and reconstruction of the clonal architecture [77,78].

CNA inference from single-cell gene expression data revealed that clonal evolution accompanies transcriptional heterogeneity in multiple cancer types, including myeloma, breast cancer, and glioblastoma [76,78,80]. In these studies, the genetic evolution potentially leading to the phenotype transition of tumor cells toward metastasis and treatment resistance could be identified. Transcriptional heterogeneity that is not recapitulated at the CNA level may indicate incomplete inference, other genomic events, and/or influences from microenvironmental factors.

### 4.2. Linking Protein and Gene Expression Clusters to the Functional Phenotype through Multi-Omic Analyses

The inference obtained from transcriptional clusters and a sparse gene expression necessitates cluster validation at the protein level, which might be provided by independent analyses such as flow cytometry or mass cytometry. Ideally, the use of antibody-derived tag (ADT) labeling cell surface proteins together with scRNA-seq [84,85] would result in the demarcation of transcriptional clusters linked to protein marker-defined subpopulations. As the ADT method is dependent on the specific binding of antibodies, combined usage with gene expression results in the generation of more reliable reference cell types [86]. An additional data type, i.e., chromatin accessibility, allows for a more refined cell type or cell state classification [87]. The direct projection of transcriptional clusters onto the functional phenotype can be achieved by single-cell clustered regularly interspaced short palindromic repeat (CRISPR) screening [87,88,89]. Moreover, spatial profiling using single-cell or spatial transcriptomics [90,91] uncovers the location of specific cell populations and their neighbors, further refining cellular identities and substantiating cellular interactions. Overall, multimodal data bridge biomarkers (from any data type) and functional phenotypes, help refine the cell type and cell state annotations, and allow for a mechanistic dissection for phenotype control.

### 4.3. Linking Single-Cell Gene Expression Data to Clinical Output through Bulk Deconvolution

Despite the high-resolution cell type or transcriptional state information, the availability of single-cell gene expression data with clinical annotations is limited when compared to the bulk tissue-level data. Therefore, once cellular characteristics are defined at a single-cell resolution, tissue-level gene expression data are used to validate the clinical significance of the findings from the single-cell data. (Table 5 and Figure 4).

First, deconvolution approaches would facilitate the inference of the cell type composition from the bulk gene expression data. Performance testing of different deconvolution methods using pseudo-bulk single-cell RNA sequencing data demonstrated that linear data transformation, an all-inclusive reference matrix, and sensible marker selection are required to achieve high concordance between the proportions of expected and calculated cell types [100]. The resolution of deconvolution approaches is confined to well-defined global cell types, whereas single-cell data allow for a fine subpopulation analysis with distinct and even transitional transcription states. The second approach involves correlation or enrichment analysis of gene expression signatures derived from a subpopulation in the single-cell data. In this approach, correlation or enrichment scores are used to portray bulk gene expression data, with more flexibility than the deconvolution approaches. Cancer cell programs [70] and immune cell programs [101,102] derived from single-cell data have been utilized to evaluate bulk datasets and are linked to clinical disease courses. As more scRNA-seq data are generated with clinical annotations, direct comparisons of fine cellular dynamics and disease courses will become available.

### 4.4. Using BCR and TCR Data for Lineage Tracing and Therapeutic Development

Single-cell RNA sequencing analysis has revealed the whole combinatorial sequences of BCR and TCR allow for the lineage tracing of clonal B/plasma cells and T cells, respectively. Lineage tracing involves direct comparison of a patient’s T cells collected from different regions, such as the blood and the normal and tumor tissues, as well as the T cells at various differentiation stages. Through simultaneous gene expression and TCR analyses, high levels of clonal expansion have been identified in cytotoxic T cells from normal tissues and exhausted/cytotoxic T cells from tumor tissues [103,104]. Clonal expansion of tumor-infiltrating T cells suggests a potential tumor reactivity that might be further propagated by immune checkpoint inhibitors. In addition to tracking T cell or antibody responses, TCR or BCR sequences may also be used for the development of novel treatments in the future. Therapeutic antibody development using this reversal approach has been adopted in non-cancer conditions such as COVID-19 infection or vaccination [105,106], where antigenic specificity can be easily tested. Finding tumor antigen specificities is a challenging task and currently limits the direct use of BCR or TCR sequences.

## 5. Conclusions

Single-cell genomics enables the fine-tuning of cancer genomics to a cellular resolution and broadens the scope of cancer cell biology to the genomic scale. For cellular and molecular dissection, various experimental and analytical methods have been developed to specify the cell types and states that matter, and to enable the molecular features that characterize heterogeneous tumor cells to be distinguished from those characterizing the support cells in the tumor microenvironment. Diverse cell populations comprising tumor tissues are not randomly recruited, as they are likely to communicate with each other and form a network that determines the cellular and the molecular landscapes of tumor tissues as an entity [107,108]. The molecular characteristics of each cellular component and its interconnections—either promoting or inhibiting tumor growth—are all points that can be leveraged during therapeutic development. Therefore, single-cell genomics and the related multi-omics technologies are exploratory tools that far exceed the scope and effectiveness of preceding bulk genomic analyses.

Beyond being an exploratory tool, single-cell genomics has limited capacity as a field technology for detecting tumor heterogeneity in precision oncology. This limitation is attributed to the current sampling breadth, which may result in a partial representation of the whole tumor. For rare CTCs and residual tumors, single-cell genomics facilitated genomic characterization by enabling partial DNA or RNA analyses. Nonetheless, capturing CTCs and residual tumors requires a very large number of cells—up to billions—for analysis or a high-fold enrichment of target populations. However, single-cell genomics technology represents a very active field of research that links single-cell data directly to the clinical decision-making process.

## Figures and Tables

**Figure 1 biomolecules-11-01161-f001:**
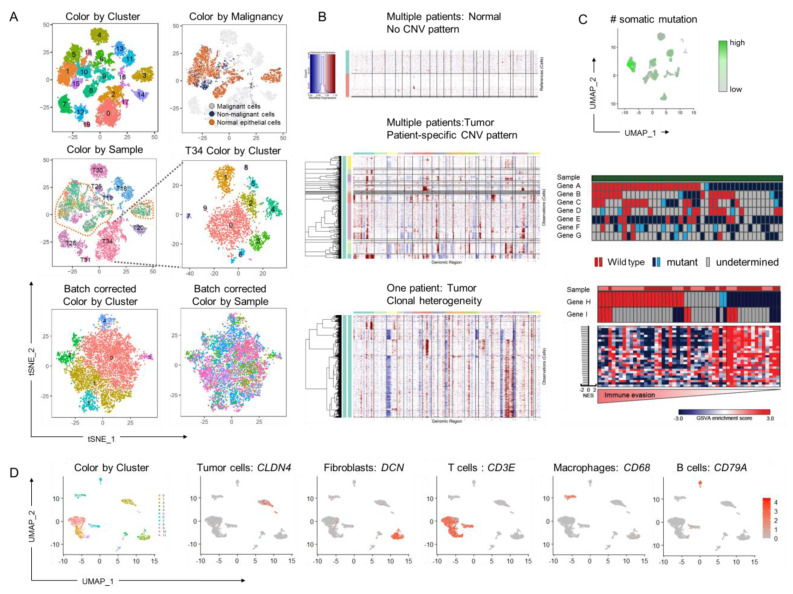
Cellular heterogeneity in cancer depicted by single cell transcriptome analysis. (**A**) Clustering analysis shows patient-specific tumor groups in multiple samples (Color by Sample) compared to the multi-patient clustering of normal and non-malignant cells (orange dotted outline). Intratumoral heterogeneity in a selected patient sample is depicted in T34 Color by Cluster. Batch correction by multi-set canonical correlation analysis (CCA) [6] clears the inter-patient cluster variation originating from both technical and biological sources (batch corrected tumor cell clusters: Color by Cluster and Color by Sample). (**B**) Copy number inference from gene expression data shows patient-specific patterns for tumor cells and intratumoral heterogeneity within a selected patient sample. (**C**) Somatic mutation calls separate a tumor cell cluster (upper UMAP) and demonstrate intratumoral heterogeneity in individual cells (lower heatmaps). (**D**) Diverse stromal and immune cell types in the tumor microenvironment. Plots are arbitrarily generated to demonstrate the depicted features.

**Figure 2 biomolecules-11-01161-f002:**
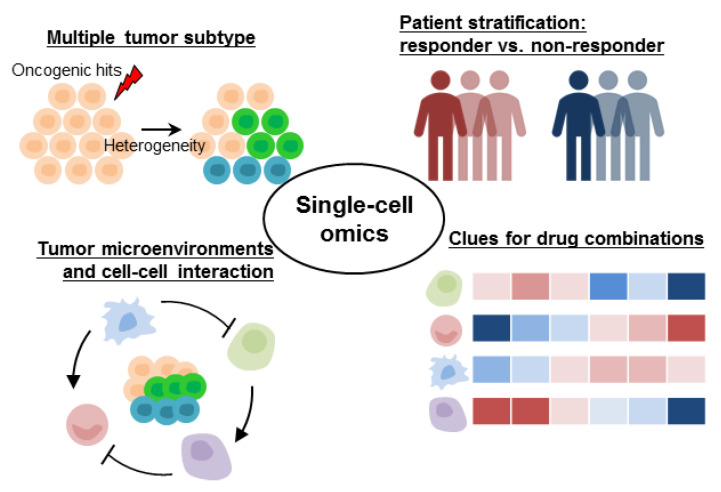
Gene expression heterogeneity in cancer addressed by scRNA-seq.

**Figure 3 biomolecules-11-01161-f003:**
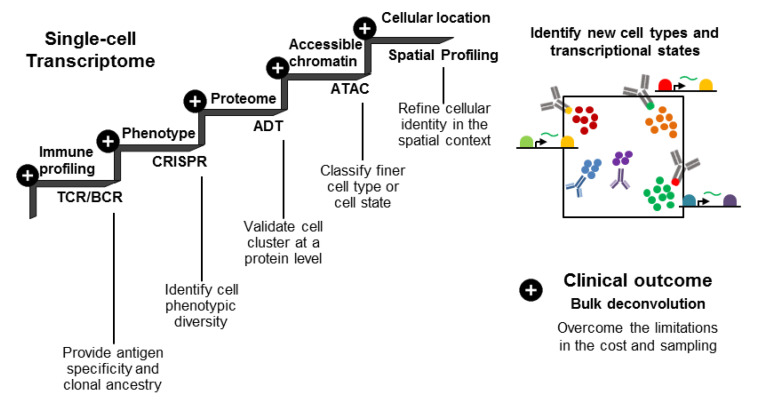
Integrative approaches using single-cell multi-modal omics.

**Figure 4 biomolecules-11-01161-f004:**
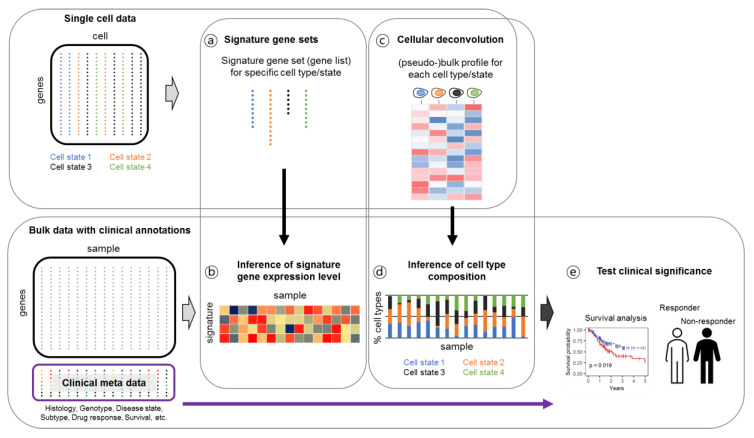
Strategies linking single cell gene expression analysis and bulk-level clinical data.

**Table 1 biomolecules-11-01161-t001:** Cancer genomics implemented in the clinical decision process.

Genomic Features Purpose	Representative Test Target Tumor	Test Platform Coverage	Diagnostic Use
Genomic DNA (+RNA in Oncomine Dx): DNA point mutations, gene amplification, translocation, TMB, HRD Molecular targeted therapy Response prediction	FoundationOne CDx Solid tumors	NGS panel 324 genes	FDA approved 16 CDx indications
	Oncomine Dx NSCLC	NGS panel 23 genes from DNA and fusions in ROS1 and RET from RNA	FDA approved 4 genes CDx in NSCLC
	MSK-IMPACT Solid tumors	NGS panel 468 genes	FDA approved No indication
RNA Gene expression Prognostic Predictive	Mammaprint Breast cancer	Microarray 70 genes	FDA approved NCCN guidelines
	Prosigna (Pam50) Breast cancer	NanoString nCounter 58 genes	FDA approved NCCN guidelines
	Oncotype Dx Breast cancer	RT-PCR 21 genes	NCCN guidelines
	Breast Cancer Index Breast cancer	RT-PCR 7 genes	NCCN guidelines
	EndoPredict test Breast cancer	RT-PCR 12 genes	NCCN guidelines

TMB, tumor mutational burden; HRD, homologous repair deficiency; CDx, companion diagnostics; NCCN, national comprehensive cancer network.

**Table 2 biomolecules-11-01161-t002:** Single-cell studies dissecting tumor signatures and specific microenvironment.

Tumor Type	Technology	Main Findings	References
NSCLC	scRNA-seq	A cancer cell subtype deviating from the normal differentiation trajectory is associated with poor survival in lung adenocarcinoma. Metastatic lung cancer reveals pro-tumoral and immunosuppressive microenvironment along with expansion of monocyte-derived macrophages and dendritic cells, and exhausted T-cells.	[45]
NSCLC	scRNA-seq	Heterogenous epithelial cell types spread along cancer developmental trajectories. Rare cell types such as follicular dendritic cells and T helper 17 cells are identified in the advanced NSCLC. cDC2 displayed Langerin (CD207) is associated with clinical outcomes in lung adenocarcinoma.	[46]
Breast cancer	snDNA-seq	Aneuploid rearrangements burst early in tumor evolution. Point mutations evolve gradually to produce extensive clonal diversity.	[47]
Breast cancer	scDNA-seq; scRNA-seq	Subclones associated with chemoresistance are pre-existing in the tumor.	[48]
Colon cancer	scRNA-seq	The classification according to consensus molecular subtypes reflects both the tumor and its associated microenvironment.	[49]
Ovarian cancer	scRNA-seq	Heterogeneous copy number alterations explain intra- and inter-patient variation of malignant cells. Diverse subsets of cancer-associated fibroblasts and macrophages show inter-patient variability.	[50]
Glioblastoma	scRNA-seq	Four cellular states of malignant cells are identified in glioblastoma, which are influenced by their associated microenvironment and exhibit plasticity.	[51]
Glioblastoma	scRNA-seq	Glioblastoma subtypes which have variable expression in diverse transcriptional programs and prognostic implications are identified.	[1]
Head and neck cancer	scRNA-seq	Head and neck squamous cell carcinoma (HNSCC) subtype vary in cell cycle, stress, hypoxia, and epithelial differentiation. A partial epithelial-to-mesenchymal transition (p-EMT), one of HNSCC subtypes, is a predictor of metastasis and progression.	[52]
Melanoma	scRNA-seq	Diverse transcriptional heterogeneity across malignant cells within a tumor is related to the cell cycle, spatial context, and a drug-resistance program. Malignant cells with elevated levels of AXL and MITF coexist in melanoma.	[2]

**Table 3 biomolecules-11-01161-t003:** Single-cell studies identifying immune cells associated with response to immune checkpoint inhibitor.

Tumor Type	Sample Source	Technology	Cell Subset	Association	References
Melanoma	TILs	Flow cytometry	PD-1 hi CTLA-4 hi CD8+ T cell	Relative abundance of PD-1hiCTLA-4hi cells within the tumor-infiltrating CD8+ T cell subset correlates with response to checkpoint inhibitors and survival.	[58]
Melanoma	PBMCs TILs	Flow cytometry	Tex cell	The invigoration of circulating exhausted-phenotype CD8+ T (Tex) cells in relation to tumor burden correlates with clinical response to PD-1 blockade.	[59]
NSCLC	PBMCs	Flow cytometry	PD-1+ CD8+ T cell	PD-1+ CD8+ T cell in peripheral blood correlates with clinical benefit after receiving PD-1-targeted therapies.	[60]
Melanoma	TILs	Mass cytometry; RNA-seq	Tex cell; ICOS+ Th1-like CD4 effector cell	Anti-PD-1 induces the expansion of tumor-infiltrating exhausted-like CD8+ T cells (Tex), but anti-CTLA-4 induces the expansion of the ICOS+ Th1-like CD4 effector cells in addition to Tex cells.	[61]
NSCLC	TILs	Flow cytometry; RNA-seq	PD-1+ CD8+ T cell	The presence of CD8+ T lymphocyte with high PD-1 expression is strongly predictive for response to PD-1 blockade and better survival.	[62]
Melanoma	tumor	scRNA-seq; ATAC-seq	TCF7+ CD8+ T cell	The presence of TCF7+ CD8+ T cells predicts response to checkpoint inhibitors and positive clinical outcome.	[63]
Melanoma	tumor	RNA-seq; Multiplex IHC; CyTOF	EOMES+ CD69+ CD45RO+ effector memory T cell	EOMES+CD69+CD45RO+ effector memory T cell phenotype is associated with response to checkpoint inhibitors and longer survival.	[64]
Melanoma	TILs PBMCs	Flow cytometry; RNA-seq	Tex cell	The accumulation of exhausted CD8 T (Tex) cells in the tumor was associated with clinical benefit and response to anti-PD-1.	[65]
Melanoma; Head&Neck; Breast; Urothelial; Renal; Lung; Colorectal cancer	tumor TILs	RNA-seq; scRNA-seq	CXCL9/CXCL13 CD8+ T cell	Clonal TMB and CXCL9/CXCL10 expression in tumor-infiltrating CD8+ T cells predicts response to checkpoint blockade.	[66]
Renal cell carcinoma	tumor lymph node PBMCs	scRNA-seq	CD8A+ tissue-resident T cell	The abundance of CD8A+ tissue-resident T cells is associated with response to immune checkpoint blockade and improved survival.	[67]
Gastric cancer	tumor	scRNA-seq	PD-1+ CD8+ T cell	The abundance of PD-1+ CD8+ T cells correlate with response and longer survival to PD-1 blockade.	[68]

**Table 4 biomolecules-11-01161-t004:** Genetic features assessed by scRNA-seq.

Genetic Feature	Method	Tool Name	References
CNA	chromosomal gene expression pattern	inferCNV	https://github.com/broadinstitute/inferCNV, accessed on 15 July 2021
		CopyKAT	[76]
		CaSpER	[77]
		HoneyBADGER	[78]
		scCNAutils	[79]
		CONICS	[80]
Mutation	targeted genotyping of cDNA	GoT	[81]
	statistical method	SCMut	[82]
		DENDRO	[83]

**Table 5 biomolecules-11-01161-t005:** Tumor studies using scRNA-seq data for bulk RNA-seq deconvolution.

Tumor Type	Deconvolution Approaches	Main Findings	References
Signature gene sets			
Triple negative breast cancer	Inference of tissue-resident memory T (Trm) gene signature expression level	Association with improved patient survival rate in early-stage TNBC	[92]
Melanoma	Inference of T cell infiltration levels and malignant genes correlated with T cell infiltration	Genes associated with T cell exclusion program	[70]
Nasopharyngeal carcinoma	Correlations of 17 immune subtype signatures with clinicopathological features	Immune subsets significantly associated with better survival outcomes	[93]
Gastric cancer	Inference of M2-like TAM level	Improved prediction of poor survival by high M2 signature	[94]
Cellular deconvolution			
Melanoma	(CIBERSORTx) Deconvolution into major cell types	(1) Cellular signatures within malignant cells and CAFs according to BRAF or NRAS mutation status (2) CD8 T cell signatures associated with anti-PD1 or anti-CTLA4 therapy response	[95]
Adrenal Neuroblastoma	(MuSiC) Deconvolution into developing cell populations	(1) Three tumor groups; Undifferentiated chromaffin cell (UCHC)-like, Differentiated chromaffin cell (DCHC)-like, and EMT NCC-like (2) EMT NCC-like tumors had the worst survival rate, while DCHC-like tumors had the better survival rate	[96]
Bladder cancer	(CIBERSORTx) Deconvolution into fine subsets of cell types	Accumulation of a CAF subtype associated with poor overall survival	[97]
Head and Neck Squamous Cell Carcinoma	(CIBERSORTx & MuSiC) Deconvolution into T cell subpopulations	High proportions of regulatory T cells as the major contributor of improved survival rate	[98]
Renal cell carcinoma	(CIBERSORTx) Infererence of CXCL10-Hi TAM and CD8+ T cell fraction wthin on-ICB T cells	Positive correlation between CXCL10-Hi TAM fraction and the amount of interferon gamma expressed by CD8	[99]

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
