# Peer review of "Clinical Perspectives of Single-Cell RNA Sequencing"

_biomolecules, 2021, doi:10.3390/biom11081161_

Round 1
Reviewer 1 Report
This review by Kim et al is a very nice summary of scRNA-seq and its relevance in clinical context. The manuscript is informative and carefully prepared.
Two minor comments:
- I would avoid using the term 'MRD tumor'. 'Residual tumor' or 'residual tumor load' would be better. Instead of 'minimal', 'measurable' could also be used since this term is becoming more and more popular when someone is talking about MRD. (the residual tumor is not always minimal and anyway, who knows what minimal is....but if we detect something, its measurable)
- The authors could include a couple of sentences about spatial transciptomics, even if it does not reach single-cell resolution, yet. That level will also be achieved soon and this aspect would read nice as an outlook.
Reviewer 2 Report
This manuscript is a comprehensive review of the status quo of single-cell RNA sequencing technology applications in the cancer discovery, prognostics, and treatment. The literature summary is thorough. Overall it provides a nice overview of this field and could be very helpful for the research community.
I have a few comments:
1) Figure 1A. There are a lot of sample-specific cluster in the TSNE plots, which are usually caused by batch effect, and in data analysis these effects are normally adjusted. So an ideal final TSNE/UMAP would show cells clustered by cell type and disease status, instead of by sample/individual. It would be great if the author can come up with a better illustration.
2) Could the author clarify more on “Exhausted phenotype T cells are clearly the targets of ICT and are associated with a positive response in melanoma, NSCLC, and basal cell carcinoma”? Normally T cell exhaustion indicates unfavourable outcomes, and the claim in the manuscript seems contradictory.
3) In table 3 there is a typo. “PMBCs” should be PBMCs“”.
